# External validation of cardiac arrest-specific prognostication scores developed for early prognosis estimation after out-of-hospital cardiac arrest in a Korean multicenter cohort

**Wan Young Heo[1], Yong Hun Jung[1,2], Hyoung Youn Lee[3], Kyung Woon Jeung[1,2]\*, Byung Kook Lee[1,2], Chun Song Youn[4], Seung Pill Choi[5], Kyu Nam Park[4], Yong Il Min[1,2], on behalf of the Korean Hypothermia Network Investigators[¶]**

1 Department of Emergency Medicine, Chonnam National University Hospital, Gwangju, Republic of Korea, 2 Department of Emergency Medicine, Chonnam National University Medical School, Gwangju, Republic of Korea, 3 Trauma Center, Chonnam National University Hospital, Gwangju, Republic of Korea, 4 Department of Emergency Medicine, Seoul St. Mary's Hospital, College of Medicine, The Catholic University of Korea, Seoul, Republic of Korea, 5 Department of Emergency Medicine, Eunpyeong St. Mary's Hospital, College of Medicine, The Catholic University of Korea, Seoul, Republic of Korea

¶ Membership of the Korean Hypothermia Network Investigators is listed in the Acknowledgments.
\* neoneti@hanmail.net

**Data Availability Statement:** All relevant data are within the paper and its Supporting information files.

## Abstract

We evaluated the performance of cardiac arrest-specific prognostication scores developed for outcome prediction in the early hours after out-of-hospital cardiac arrest (OHCA) in predicting long-term outcomes using independent data. The following scores were calculated for 1,163 OHCA patients who were treated with targeted temperature management (TTM) at 21 hospitals in South Korea: OHCA, cardiac arrest hospital prognosis (CAHP), C-GRApH (named on the basis of its variables), TTM risk, 5-R, NULL-PLEASE (named on the basis of its variables), Serbian quality of life long-term (SR-QOLI), cardiac arrest survival, revised post-cardiac arrest syndrome for therapeutic hypothermia (rCAST), Polish hypothermia registry (PHR) risk, and PROgnostication using LOGistic regression model for Unselected adult cardiac arrest patients in the Early stages (PROLOGUE) scores and prediction score by Aschauer et al. Their accuracies in predicting poor outcome at 6 months after OHCA were determined using the area under the receiver operating characteristic curve (AUC) and calibration belt. In the complete-case analyses, the PROLOGUE score showed the highest AUC (0.923; 95% confidence interval [CI], 0.904–0.941), whereas the SR-QOLI score had the lowest AUC (0.749; 95% CI, 0.711–0.786). The discrimination performances were similar in the analyses after multiple imputation. The PROLOGUE, TTM risk, CAHP, NULL-PLEASE, 5-R, and cardiac arrest survival scores were well calibrated. The rCAST and PHR risk scores showed acceptable overall calibration, although they showed miscalibration under the 80% CI level at extreme prediction values. The OHCA score, C-GRApH score, prediction score by Aschauer et al., and SR-QOLI score showed significant miscalibration in both complete-case (P = 0.026, 0.013, 0.005, and < 0.001, respectively) and multiple-imputation analyses (P = 0.007, 0.018, < 0.001, and < 0.001, respectively). In

**Funding:** This study was supported by a grant (BCRI21040) from the Chonnam National University Hospital Biomedical Research Institute (Recipient: KWJ). The funders had no role in the study design, data collection and analysis, decision to publish, or preparation of the manuscript.

**Competing interests:** The authors have declared that no competing interests exist.

conclusion, the discrimination performances of the prognostication scores were all acceptable, but some showed significant miscalibration.

## Introduction

Out-of-hospital cardiac arrest (OHCA) remains the leading cause of mortality and disability worldwide [1, 2]. Most of the patients resuscitated from OHCA eventually die in hospital or develop severe neurologic sequelae; only 10%–30% survive with good neurologic outcome [2, 3]. Current guidelines recommend delaying neurologic prognosis estimation in comatose cardiac arrest patients until at least 72 h after return of spontaneous circulation (ROSC) [4]. However, there is a need for an accurate prognostic tool useful during the early hours after OHCA. In the case of comatose OHCA patients, families desire precise information on the neurologic prognoses as early as possible. Treating physicians often have to make critical decisions regarding the use of costly and resource-intensive therapies, such as extracorporeal membrane oxygenation (ECMO), in the early stages of post-cardiac arrest care, when the patients' neurologic prognoses are uncertain.

Several cardiac arrest-specific prognostication scores for use in the early hours after OHCA have been developed from retrospective or prospective analyses of OHCA data [5–16]. These scores have several limitations that must be addressed to render them useful in clinical practice. A risk prediction score derived from one study population may not be accurate in other populations. Thus, external validations in various patient populations are required to enable widespread reliance on a risk prediction score, but few such scores have undergone any external validation using independent data; where this has been done, it was usually limited to retrospective analyses of discrimination performance [7, 9, 17–22]. Most of the scores are intended to predict short-term outcomes, such as survival to hospital discharge or neurologic outcome at hospital discharge [5–8, 10–12, 14–16], and have not been evaluated as a means to predict long-term outcomes. Targeted temperature management (TTM) is now the standard treatment for comatose OHCA patients. However, several scores were developed before the widespread use of TTM or derived from studies that included OHCA patients irrespective of whether they had undergone TTM [5–7, 10, 12–14].

To address these limitations, we sought to evaluate the performance of cardiac arrest-specific prognostication scores developed for outcome prediction in the early hours after OHCA in predicting long-term outcomes, using independent data from a multicenter registry of comatose OHCA patients who underwent TTM. We hypothesized that the scores would accurately predict long-term outcomes in an independent cohort of OHCA patients who underwent TTM.

## Materials and methods

### Study design and setting

This study conformed to the principles outlined in the Declaration of Helsinki. It was a retrospective analysis of data from the Korean Hypothermia Network prospective (KORHN-pro) registry, which enrolled adult OHCA patients treated with TTM at 22 teaching hospitals in the Republic of Korea [3]. In brief, a principal investigator at each participating hospital reviewed the medical records of patients who were eligible for registry enrollment and collected their demographic, prehospital resuscitation, in-hospital treatment, and outcomes data in an anonymous fashion using a web-based case report form based on the Utstein Resuscitation Registry

Templates [23]. Data quality was assured by five clinical research associates who queried any concerns with the investigators, and a data manager with final responsibility for determining data acceptability. The study design and registry protocol were approved by the institutional review board of all participating hospitals, including the Chonnam National University Hospital Institutional Review Board (CNUH-2015-164) and registered at the International Clinical Trials Registry Platform (ClinicalTrials.gov identifier: NCT02827422). Written informed consent was obtained from the legal surrogates of all patients enrolled in the registry.

## Study population

The KORHN-pro registry included all adult ($\geq$ 18 years) unconscious (Glasgow Coma Scale [GCS] score $<$ 8) OHCA survivors treated with TTM at participating hospitals between October 2015 and December 2018, except those with the following conditions: OHCA associated with hemorrhagic or ischemic stroke; poor pre-arrest neurologic status (cerebral performance category [CPC] of 3 or 4); body temperature $<$ 30˚C on admission; pre-arrest do-not-resuscitate order; or known terminal illness leading to life expectancy $<$ 6 months. One of the scores included in this study (PROLOGUE [PROgnostication using LOGistic regression model for Unselected adult cardiac arrest patients in the Early stages]) is developed using data from one of the participating hospitals [7]. Thus, patients enrolled from this center were excluded from this study, as were patients without data on outcomes at 6 months. The patients included in the registry were managed according to the treatment protocols of each hospital.

## Variables

Data on the following variables were obtained for each patient: age, sex, hospital, pre-existing chronic diseases (coronary artery disease, heart failure, arrhythmia, cerebrovascular accident [CVA], neurologic disease other than CVA, diabetes, hypertension, pulmonary disease, chronic kidney disease, liver cirrhosis, and malignancy), patient location at the time of cardiac arrest, presence of a witness to the collapse, bystander cardiopulmonary resuscitation (CPR), first monitored rhythm, no-flow duration, low-flow duration, time to ROSC, dose of epinephrine given during CPR, etiology of cardiac arrest, circulatory status on emergency department arrival (prehospital ROSC), GCS motor score and pupillary light reflex obtained before intensive care unit (ICU) admission, initial laboratory parameters after ROSC (lactate, arterial pH, partial pressure of arterial oxygen [$PaO_2$], partial pressure of arterial carbon dioxide [$PaCO_2$], potassium, phosphate, creatinine, glucose, and hemoglobin), duration and target temperature of TTM, Sequential Organ Failure Assessment score on the first day after hospital admission, occurrence of rearrest before ICU admission, critical care interventions implemented during hospitalization (coronary angiography and ECMO), length of hospital stay, and CPC at 6 months after OHCA. No-flow and low-flow durations were defined as the time interval from collapse to first CPR attempt and the time interval from first CPR attempt to ROSC, respectively. Time to ROSC was defined as the time interval from collapse to ROSC. CPC at 6 months after OHCA was evaluated through in-person or telephone interviews conducted by medical staff at each center who were blinded to patient data. A CPC of 1 or 2 was defined as a good outcome and a CPC of 3–5 as a poor one (primary outcome). After literature review, the following cardiac arrest-specific prognostication scores were selected based on availability of the data required for score calculation and were calculated using the formulas presented in the original publications. The scores were as follows: OHCA [5]; cardiac arrest hospital prognosis (CAHP) [6]; PROLOGUE [7]; C-GRApH [8], named on the basis of its variables; TTM risk [9]; prediction score by Aschauer et al. [10]; 5-R [11]; NULL-PLEASE [12], named on the basis of its variables; Serbian quality of life long-term (SR-QOLl) [13]; cardiac arrest survival [14];

revised post-cardiac arrest syndrome for therapeutic hypothermia (rCAST) [15]; and Polish hypothermia registry (PHR) risk [16]. The characteristics of these scores are summarized in Table 1. A greater risk of poor outcome is indicated by lower scores for the 5-R and SR-QOLl scores, but otherwise by higher scores.

## Statistical analysis

Data analysis and reporting were performed in accordance with the Transparent Reporting of a multivariable prediction model for Individual Prognosis or Diagnosis statement [24]. The sample size of this study far exceeded the suggested minimum sample size for external validation studies of multivariable prediction models [25, 26]. Statistical analyses were conducted using T&F programme version 3.0 (YooJin BioSoft, Goyang, Republic of Korea) and R language version 4.0.3 (R Foundation for Statistical Computing, Vienna, Austria). Continuous variables are presented by medians with interquartile ranges, unless otherwise specified. Categorical variables are expressed as numbers of cases with percentages. Comparisons between two independent groups were performed using the Mann–Whitney U test for continuous variables and the chi-square test with continuity correction for categorical variables. To determine the association of each prognostication score with the primary outcome, binary logistic regression analyses were performed after dividing the patients into two groups according to the optimal cut-off for each score. The discrimination abilities of the prognostication scores were assessed using receiver operating characteristic (ROC) analysis, and quantified with area under the ROC curve (AUC). The AUC values were compared in a pairwise manner using the method of DeLong et al. [27]. For each score, sensitivity, specificity, positive predictive value, negative predictive value, and accuracy were calculated for the optimal cut-off, determined using the Youden index. The calibration performances of the prognostication scores were assessed using the calibration belt [28, 29]. To allow for comparisons between scores, score performances were initially evaluated for patients for whom all 12 score values were calculable. To evaluate the robustness of the results, missing values of the variables required for the calculation of the prognostication scores were imputed using the MICE package in R, and the performances of the prognostication scores were reassessed. Statistical significance was indicated by a two-sided P-value of $< 0.05$.

## Results

A total of 1,373 adult OHCA patients treated with TTM were enrolled in the KORHN-pro registry. Among these, 187 who were enrolled from the hospital involved in the development of PROLOGUE and 23 without data on CPC at 6 months after OHCA were excluded from this study, leaving 1,163 included patients (Fig 1). These were mostly male (70.9%), with a median age of 58.3 years old (46.8–69.9). The majority of the patients had a witnessed cardiac arrest (71.6%), received bystander CPR (63.1%), and presented with a non-shockable initial cardiac arrest rhythm (63.2%). The no-flow duration, low-flow duration, and time to ROSC were 1.0 (0.0–6.0), 25.0 (14.0–38.0), and 30.0 (18.0–43.0) min, respectively. The cardiac arrest was cardiac in origin in 714 (61.4%) patients. Four hundred (34.4%) patients underwent coronary angiography, and 57 (4.9%) received ECMO during hospitalization. Of the included patients, 357 (30.7%) had a good outcome 6 months after OHCA, while the remaining 806 (69.3%) patients had a poor outcome. The clinical and laboratory characteristics of patients, stratified by outcomes at 6 months after OHCA, are summarized in Table 2. As shown in Table 2, all 12 of the prognostication scores in the present study were significantly associated with the primary outcome (all P < 0.001).

**Table 1. Details of the prognostication scores included in the present study.**

| Prognostication score | Predicted outcome | Components of score | Population used for development | Discriminatory ability in the original publication |
|---|---|---|---|---|
| OHCA score [5] | CPC 3–5 at hospital discharge | First monitored rhythm; no-flow duration; low-flow duration; creatinine; lactate | 130 adult OHCA survivors admitted to a French ICU between 1999 and 2003 | Derivation cohort: AUC 0.82 (95% CI, 0.70–0.95) Validation cohort: AUC 0.88 (95% CI, 0.82–0.94) |
| CAHP score [6] | CPC 3–5 at hospital discharge | Age; location of cardiac arrest; first monitored rhythm; no-flow duration; low-flow duration; pH; epinephrine dose | 819 OHCA survivors in a multicenter registry in Paris and suburbs between 2011 and 2012 | Derivation cohort: AUC 0.93 (95% CI, 0.91–0.95) Validation cohort: AUC 0.85 (95% CI, 0.82–0.91), AUC 0.91 (95% CI, 0.88–0.93) |
| PROLOGUE [7] | CPC 3–5 at hospital discharge | Presence of a witness on collapse; potassium; lactate; epinephrine dose; low-flow duration; hemoglobin; creatinine; phosphate; first monitored rhythm; pupillary light reflex; age; GCS motor score | 671 adult cardiac arrest survivors admitted to a university hospital in South Korea between 2014 and 2016 | Derivation cohort: AUC 0.940 (95% CI, 0.923–0.956) Internal validation: AUC 0.930 (95% CI, 0.912–0.949) Validation cohort: AUC 0.942 (95% CI, 0.917–0.968) |
| C-GRApH score [8] | CPC 1–2 at hospital discharge | Pre-existing coronary artery disease; glucose; first monitored rhythm; age; pH | 122 adult OHCA survivors treated with TTM at a hospital in the USA between 2008 and 2012 | Derivation cohort: c-statistic 0.818 (95% CI, 0.737–0.899) Validation cohort: c-statistic 0.814 (95% CI, 0.759–0.869) |
| TTM risk score [9] | CPC 3–5 at 6 months after OHCA | Age; location of cardiac arrest; first monitored rhythm; no-flow duration; low-flow duration; epinephrine dose; GCS motor score; PaCO$_2$ | 933 OHCA survivors included in the TTM trial | Derivation cohort: AUC 0.842 (95% CI, 0.840–0.845) Internal validation: AUC 0.818 (95% CI, 0.816–0.821) |
| Prediction score by Aschauer et al. [10] | Survival at 30 days after OHCA | Time to ROSC; age; first monitored rhythm; epinephrine dose | 1,242 OHCA survivors admitted to a university hospital in Austria between 2000 and 2012 | Validation cohort: AUC 0.810 |
| 5-R score [11] | CPC 1–2 at hospital discharge | No-flow duration; first monitored rhythm; time to ROSC; rearrest; pupillary light reflex | 66 OHCA survivors treated with TTM at a hospital in Japan between 2006 and 2011 | Derivation cohort: AUC 0.95 (95% CI, 0.89–10) |
| NULL-PLEASE score [12] | In-hospital mortality | First monitored rhythm; presence of a witness on collapse; bystander CPR; low-flow duration; pH; lactate; pre-existing chronic kidney disease; age; circulatory status on emergency department arrival; etiology of cardiac arrest | 56 OHCA survivors admitted to an ICU in the UK | AUC: not available No patient with a NULL-PLEASE score of > 6 survived to hospital discharge |
| SR-QOLl score [13] | Survival at 1 year after hospital discharge | Bystander CPR; first monitored rhythm; presence of a witness on collapse; no-flow duration; etiology of cardiac arrest; age | 591 adult patients who experienced OHCA in four Serbian cities between 2007 and 2008 | Derivation cohort: AUC 0.913 ± 0.026 |
| Cardiac arrest survival score [14] | In-hospital mortality | Age; presence of a witness on collapse; location of cardiac arrest; bystander CPR; first monitored rhythm | 2,685 adult OHCA survivors included in a large metropolitan cardiac arrest registry in USA between 2007 and 2015 | Derivation cohort: AUC 0.7172 Validation cohort: AUC 0.7081 |
| rCAST score [15] | CPC 3–5 at 30 and 90 days after OHCA | First monitored rhythm; presence of a witness on collapse; time to ROSC; pH; lactate; GCS motor score | 460 adult OHCA survivors who were treated with TTM and were included in a multicenter registry in Japan between 2014 and 2015 | Derivation cohort: AUC 0.892 and 0.895 for CPC 3–5 at 30 and 90 days after OHCA, respectively |
| PHR risk score [16] | In-hospital mortality | Age; no-flow duration; time to ROSC; location of cardiac arrest (out-of-hospital versus in-hospital); presence of a witness on collapse; first monitored rhythm | 376 cardiac arrest survivors who were treated with TTM and included in a Polish multicenter registry between 2012 and 2016 | Derivation cohort: AUC 0.742 |

OHCA, out-of-hospital cardiac arrest; CPC, cerebral performance category; ICU, intensive care unit; AUC, area under the receiver operating characteristic curve; CI, confidence interval; CAHP, cardiac arrest hospital prognosis; PROLOGUE, PROgnostication using LOGistic regression model for Unselected adult cardiac arrest patients in the Early stages; TTM, targeted temperature management; GCS, Glasgow Coma Scale; PaCO$_2$, partial pressure of arterial carbon dioxide; ROSC, restoration of spontaneous circulation; CPR, cardiopulmonary resuscitation; SR-QOLl, Serbian quality of life long-term; rCAST, revised post-cardiac arrest syndrome for therapeutic hypothermia; PHR, Polish hypothermia registry.

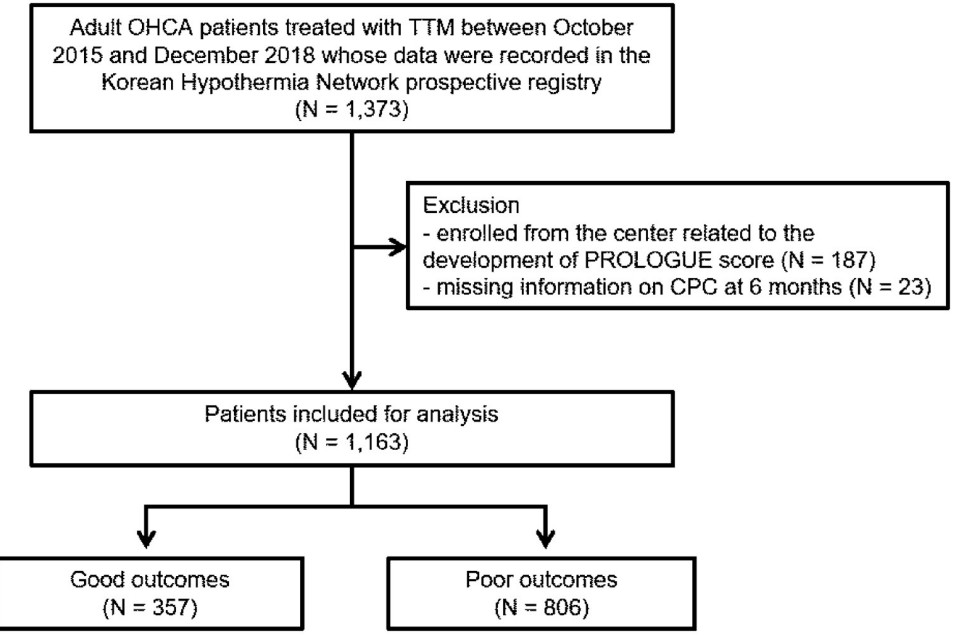

**Fig 1. Flow chart describing the patient selection process.** OHCA, out-of-hospital cardiac arrest; TTM, targeted temperature management; PROLOGUE, PROgnostication using LOGistic regression model for Unselected adult cardiac arrest patients in the Early stages; CPC, cerebral performance category.

## Prognostic performances of the scores

There was a total of 804 patients for whom all 12 prognostication scores were calculable, of whom 241 (30.0%) had a good outcome and 563 (70.0%) had a poor outcome. In binary logistic regression analyses examining the association between scores above the optimal cut-off (below the optimal cut-off for the 5-R and SR-QOLl scores) and the risk of poor outcome at 6 months after OHCA for each score (Fig 2), the odds ratios ranged from 6.813 (C-GRApH score) to 32.143 (PROLOGUE). The discrimination abilities of the prognostication scores in these patients are shown in Fig 3 and Table 3. All scores could predict poor outcome at 6 months after OHCA with statistical significance (all P < 0.001). PROLOGUE showed the highest AUC (0.923; 95% confidence interval [CI], 0.904–0.941), whereas the SR-QOLl score had the lowest AUC (0.749; 95% CI, 0.711–0.786). All scores showed similar AUC in the analyses after multiple imputation (Table 4). Table 5 shows sensitivity, specificity, positive and negative predictive values, and accuracy for different cut-offs. The results of pairwise comparisons of the ROC curves are summarized in Table 6.

The calibration performances of the prognostication scores in the 804 patients are shown in Fig 4. Calibration belts for the PROLOGUE, TTM risk, CAHP, NULL-PLEASE, 5-R, and cardiac arrest survival scores contained bisecting lines (representing perfect calibration) across the entire range of predictions. For the rCAST and PHR risk scores, the 80% CI boundaries of the calibration belt did not contain bisecting lines at extreme predicted probability values, although such lines were present in the 95% CI boundaries of calibration belts across the entire range of predictions (P = 0.060 and 0.114, respectively). Calibration belts for the prediction score by Aschauer et al. (P = 0.005), OHCA score (P = 0.026), C-GRApH score (P = 0.013), and SR-QOLl score (P < 0.001) significantly deviated from the bisecting line. This was also true in the analyses following inclusion of imputed data (prediction score by

**Table 2. Characteristics of patients stratified by outcomes at 6 months after cardiac arrest.**

| Variable | Good outcome (N = 357) | Poor outcome (N = 806) | P value |
|---|---|---|---|
| Male sex, N (%) | 280 (78.4) | 545 (67.6) | <0.001 |
| Age, years, median (IQR) | 53.9 (43.9–61.6) | 61.2 (48.7–72.9) | <0.001 |
| Witnessed collapse, N (%) | 305 (85.7)[a] | 518 (65.3)[b] | <0.001 |
| Bystander CPR, N (%) | 246 (70.1)[c] | 475 (60.0)[d] | 0.001 |
| First monitored rhythm | | | <0.001 |
| Shockable, N (%) | 265 (77.3)[e] | 151 (19.2)[f] | |
| Non-shockable, N (%) | 78 (22.7) | 637 (80.8) | |
| Comorbidities | | | |
| Coronary artery disease, N (%) | 58 (16.2) | 92 (11.4) | 0.030 |
| Arrhythmia, N (%) | 20 (5.6) | 40 (5.0) | 0.756 |
| Heart failure, N (%) | 16 (4.5) | 40 (5.0) | 0.838 |
| CVA, N (%) | 11 (3.1) | 47 (5.8) | 0.066 |
| Hypertension, N (%) | 105 (29.4) | 299 (37.1) | 0.013 |
| Diabetes, N (%) | 56 (15.7) | 212 (26.3) | <0.001 |
| Pulmonary disease, N (%) | 12 (3.4) | 77 (9.6) | <0.001 |
| Neurologic disease other than CVA, N (%) | 6 (1.7) | 51 (6.3) | 0.001 |
| Malignancy, N (%) | 17 (4.8) | 50 (6.2) | 0.403 |
| Chronic kidney disease, N (%) | 12 (3.4) | 76 (9.4) | <0.001 |
| Liver cirrhosis, N (%) | 1 (0.3) | 19 (2.4) | 0.023 |
| Time to ROSC, min, median (IQR) | 18 (12–26) | 35 (24–48) | <0.001 |
| No-flow duration, min, median (IQR) | 1 (0–5) | 1 (0–7) | 0.009 |
| Low-flow duration, min, median (IQR) | 15 (9–24) | 31 (21–42) | <0.001 |
| Epinephrine dose given during CPR, mg, median (IQR) | 0 (0–1)[g] | 2 (1–4)[h] | <0.001 |
| Arrest etiology | | | <0.001 |
| Cardiac, N (%) | 312 (87.4) | 402 (49.9) | |
| Non-cardiac, N (%) | 45 (12.6) | 404 (50.1) | |
| Cardiac arrest at home, N (%) | 151 (43.5)[i] | 451 (57.3)[j] | <0.001 |
| Prehospital ROSC, N (%) | 251 (70.3) | 122 (15.1) | <0.001 |
| Rearrest before ICU admission, N (%) | 8 (2.2) | 56 (6.9) | 0.002 |
| GCS motor score before ICU admission, median (IQR) | 2 (1–4)[k] | 1 (1–1)[l] | <0.001 |
| Reactive pupillary light reflex before ICU admission, N (%) | 264 (84.6)[m] | 313 (43.1)[n] | <0.001 |
| Initial laboratory parameters | | | |
| pH, median (IQR) | 7.23 (7.12–7.31)[o] | 7.05 (6.90–7.18)[p] | <0.001 |
| PaCO$_2$, mmHg, median (IQR) | 38.1 (32.0–46.8)[q] | 53.0 (37.5–73.6)[r] | <0.001 |
| PaO$_2$, mmHg, median (IQR) | 119.0 (81.6–218.5)[s] | 127.9 (79.3–226.3)[t] | 0.458 |
| Lactate, mmol/l, median (IQR) | 6.3 (2.1–10.6)[u] | 10.1 (5.7–13.3)[v] | <0.001 |
| Creatinine, mg/dl, median (IQR) | 1.19 (1.00–1.35)[w] | 1.31 (1.09–1.72)[x] | <0.001 |
| Potassium, mEq/l, median (IQR) | 3.8 (3.4–4.4)[y] | 4.4 (3.8–5.3)[z] | <0.001 |
| Phosphate, mg/dl, median (IQR) | 5.6 (4.0–7.1)[aa] | 7.6 (6.1–9.5)[ab] | <0.001 |
| Hemoglobin, g/dl, median (IQR) | 14.6 (13.2–15.7)[ac] | 12.7 (10.9–14.3)[ad] | <0.001 |
| Glucose, mg/dl, median (IQR) | 239 (182–295)[ae] | 266 (194–345)[af] | <0.001 |
| SOFA score on first day, median (IQR) | 9 (7–11)[ag] | 12 (10–14)[ah] | <0.001 |
| Target temperature of TTM, ˚C, median (IQR) | 33.0 (33.0–34.5) | 33.0 (33.0–34.0) | 0.542 |
| Duration of TTM, h, median (IQR) | 24 (24–24) | 24 (24–24) | 0.008 |
| ECMO, N (%) | 20 (5.6) | 37 (4.6) | 0.555 |
| Coronary angiography, N (%) | 241 (67.5) | 159 (19.7) | <0.001 |
| Cardiac arrest-specific prognostication scores | | | |

(*Continued*)

**Table 2.** (Continued)

| Variable | Good outcome (N = 357) | Poor outcome (N = 806) | P value |
|---|---|---|---|
| PROLOGUE (predicted poor outcome probability), median (IQR) | 0.224 (0.068–0.569)[ai] | 0.953 (0.829–0.985)[aj] | <0.001 |
| OHCA score, median (IQR) | 18.99 (8.34–31.09)[ak] | 42.02 (31.69–52.54)[al] | <0.001 |
| CAHP score, median (IQR) | 117.80 (99.09–149.32)[am] | 203.18 (173.33–232.58)[an] | <0.001 |
| C-GRApH score, median (IQR) | 2 (1–2)[ao] | 3 (2–3)[ap] | <0.001 |
| TTM risk score, median (IQR) | 8 (6–12)[aq] | 18 (15–22)[ar] | <0.001 |
| Prediction score by Aschauer et al., median (IQR) | 12 (7–22)[as] | 34 (26–42)[at] | <0.001 |
| 5-R score, median (IQR) | 6 (5–7)[au] | 3 (2–4)[av] | <0.001 |
| NULL-PLEASE score, median (IQR) | 2 (1–4)[aw] | 7 (5–9)[ax] | <0.001 |
| SR-QOLl score, median (IQR) | 44.0 (30.0–53.0)[ay] | 28.0 (16.0–39.5)[az] | <0.001 |
| Cardiac arrest survival score, median (IQR) | 2.5 (0–6.5)[ba] | 10.5 (8.0–14.5)[bb] | <0.001 |
| rCAST score, median (IQR) | 6.0 (3.0–9.0)[bc] | 13.0 (10.0–15.5)[bd] | <0.001 |
| PHR risk score, median (IQR) | -0.51 (-7.85–5.44)[be] | 16.86(8.06–22.44)[bf] | <0.001 |

IQR, interquartile range; CPR, cardiopulmonary resuscitation; CVA, cerebrovascular accident; ROSC, restoration of spontaneous circulation; ICU, intensive care unit; GCS, Glasgow Coma Scale; $PaCO_2$, partial pressure of arterial carbon dioxide; $PaO_2$, partial pressure of arterial oxygen; SOFA, Sequential Organ Failure Assessment; TTM, targeted temperature management; ECMO, extracorporeal membrane oxygenation; PROLOGUE, PROgnostication using LOGistic regression model for Unselected adult cardiac arrest patients in the Early stages; OHCA, out-of-hospital cardiac arrest; CAHP, cardiac arrest hospital prognosis; SR-QOLl, Serbian quality of life long-term; rCAST, revised post-cardiac arrest syndrome for therapeutic hypothermia; PHR, Polish hypothermia registry. Missing data;
[a] N = 1; [b] N = 13; [c] N = 6; [d] N = 14; [e] N = 14; [f] N = 18; [g] N = 9; [h] N = 17; [i] N = 10; [j] N = 19; [k] N = 3; [l] N = 3; [m] N = 45; [n] N = 79; [o] N = 7; [p] N = 35; [q] N = 6; [r] N = 35; [s] N = 18; [t] N = 52; [u] N = 15; [v] N = 24; [w] N = 47; [x] N = 80; [y] N = 46; [z] N = 80; [aa] N = 80; [ab] N = 162; [ac] N = 45; [ad] N = 79; [ae] N = 2; [af] N = 2; [ag] N = 14; [ah] N = 20; [ai] N = 102; [aj] N = 217; [ak] N = 66; [al] N = 117; [am] N = 39; [an] N = 85; [ao] N = 22; [ap] N = 54; [aq] N = 73; [ar] N = 145; [as] N = 23; [at] N = 32; [au] N = 58; [av] N = 97; [aw] N = 37; [ax] N = 85; [ay] N = 20; [az] N = 40; [ba] N = 29; [bb] N = 53; [bc] N = 35; [bd] N = 77; [be] N = 15; [bf] N = 30.

Aschauer et al., P < 0.001; OHCA score, P = 0.007; C-GRApH score, P = 0.018; and SR-QOLl score, P < 0.001).

## Discussion

We evaluated the performances of 12 existing prediction scores developed for early prognosis estimation after OHCA in predicting poor outcome at 6 months after cardiac arrest using independent data from a multicenter registry of comatose OHCA patients who underwent TTM. In this study, the discrimination performances of the scores were all acceptable, some even being excellent. However, some scores (prediction score by Aschauer et al., OHCA score, C-GRApH score, and SR-QOLl score) showed significant miscalibration. To the best of our knowledge, this is the largest study to evaluate the performances of multiple cardiac arrest-specific prognostication scores in an East Asian population.

Our study population differed in many aspects from the original study populations used to develop the scores included in this study. Most of the scores were derived from studies conducted in European countries or Unites States [5, 6, 8–10, 12–14, 16], where the prehospital and in-hospital care processes are quite different from Korean practice. In the patient populations used to derive the C-GRApH, TTM risk, 5-R, and PHR risk scores [8, 9, 11, 16], the proportion with initial shockable rhythm was over 85%; in contrast, this proportion was only 36.8% in our study. The proportion of witnessed arrest was 71.6% in our study population, whereas it was higher than 85% in the study populations for the CAHP score, C-GRApH score, TTM risk score, 5-R score, and prediction score by Aschauer et al. [6, 8–11]. In contrast to our study population, only 11% and 51.7% of patients were treated with TTM in the studies generating the OHCA and PROLOGUE scores, respectively [5, 7]. In addition, the primary

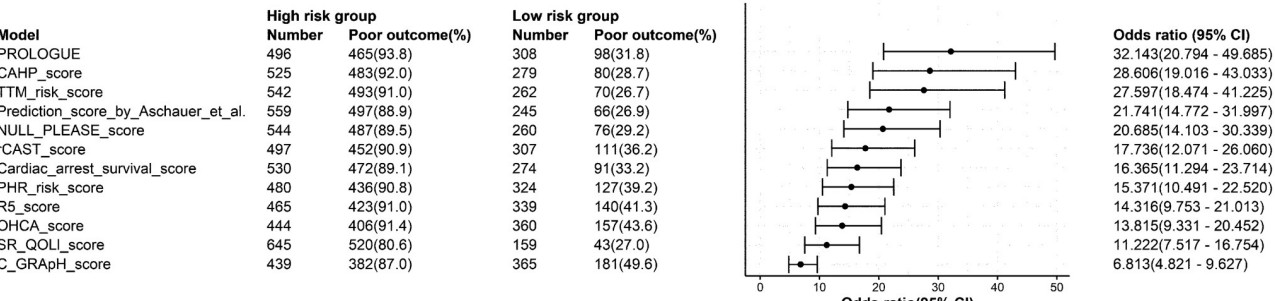

**Fig 2. Forest plot showing the association between scores above the optimal cut-off (or below the optimal cut-off for the 5-R and SR-QOLl scores) and the risk of poor outcome at 6 months after cardiac arrest.** This analysis only included the 804 patients for whom all of the 12 prognostic scores were available. The scores are displayed in descending order of odds ratio. CI, confidence interval; PROLOGUE, PROgnostication using LOGistic regression model for Unselected adult cardiac arrest patients in the Early stages; CAHP, cardiac arrest hospital prognosis; TTM, targeted temperature management; rCAST, revised post-cardiac arrest syndrome for therapeutic hypothermia; PHR, Polish hypothermia registry; OHCA, out-of-hospital cardiac arrest; SR-QOLl, Serbian quality of life long-term.

outcome of our study was poor outcome at 6 months after OHCA, whereas most of the scores were developed for prediction of outcomes at hospital discharge [5–8, 11, 12, 14, 16]. Despite these differences, the PROLOGUE, TTM risk, CAHP, NULL-PLEASE, 5-R, and cardiac arrest survival scores demonstrated satisfactory discrimination and calibration performances for predicting poor outcome at 6 months after OHCA. Although the calibration performance was not perfect, the rCAST and PHR risk scores also showed acceptable overall calibration and decent discrimination performances. These results not only support the robustness and generalizability of these scores, but also extend their applicability to the prediction of long-term outcomes.

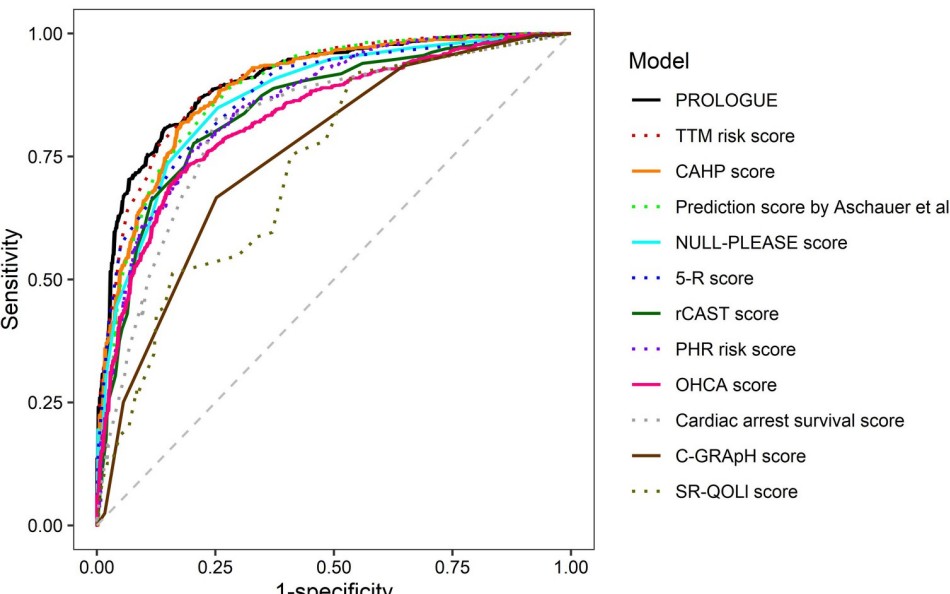

**Fig 3. Receiver operating characteristic curves of prognostication scores for predicting poor outcome at 6 months after cardiac arrest.** This analysis only included the 804 patients for whom all of the 12 prognostication scores were available. PROLOGUE, PROgnostication using LOGistic regression model for Unselected adult cardiac arrest patients in the Early stages; TTM, targeted temperature management; CAHP, cardiac arrest hospital prognosis; rCAST, revised post-cardiac arrest syndrome for therapeutic hypothermia; PHR, Polish hypothermia registry; OHCA, out-of-hospital cardiac arrest; SR-QOLl, Serbian quality of life long-term.

**Table 3. Performances of cardiac arrest-specific prognostication scores in predicting poor outcome at 6 months after cardiac arrest.**

| Prediction score | AUC (95% CI) | P value | SEN (95% CI) | SPE (95% CI) | PPV (95% CI) | NPV (95% CI) | ACC (95% CI) | Cut-off |
|---|---|---|---|---|---|---|---|---|
| PROLOGUE | 0.923 (0.904–0.941) | <0.001 | 82.6 (79.5–85.7) | 87.1 (82.9–91.4) | 93.8 (91.6–95.9) | 68.2 (63.0–73.4) | 84.0 (81.4–86.5) | 0.752 |
| TTM risk score | 0.913 (0.892–0.935) | <0.001 | 90.8 (88.4–93.2) | 73.9 (68.3–79.4) | 89.0 (86.5–91.6) | 77.4 (72.0–82.8) | 85.7 (83.3–88.1) | 13 |
| CAHP score | 0.906 (0.884–0.929) | <0.001 | 85.8 (82.9–88.7) | 82.6 (77.8–87.4) | 92.0 (89.7–94.3) | 71.3 (66.0–76.6) | 84.8 (82.3–87.3) | 156.92 |
| Prediction score by Aschauer et al. | 0.892 (0.867–0.917) | <0.001 | 88.3 (85.6–90.9) | 74.3 (68.8–79.8) | 88.9 (86.3–91.5) | 73.1 (67.5–78.6) | 84.1 (81.6–86.6) | 20 |
| NULL-PLEASE score | 0.886 (0.861–0.910) | <0.001 | 91.8 (89.6–94.1) | 65.1 (59.1–71.2) | 86.0 (83.3–88.8) | 77.3 (71.6–83.1) | 83.8 (81.3–86.4) | 5 |
| 5-R score | 0.879 (0.855–0.903) | <0.001 | 75.1 (71.6–78.7) | 82.6 (77.8–87.4) | 91.0 (88.4–93.6) | 58.7 (53.5–63.9) | 77.4 (74.5–80.3) | 4 |
| rCAST score | 0.867 (0.840–0.894) | <0.001 | 80.3 (77.0–83.6) | 81.3 (76.4–86.2) | 90.9 (88.4–93.5) | 63.8 (58.5–69.2) | 80.6 (77.9–83.3) | 10.0 |
| PHR risk score | 0.865 (0.838–0.893) | <0.001 | 77.4 (74.0–80.9) | 81.7 (76.9–86.6) | 90.8 (88.3–93.4) | 60.8 (55.5–66.1) | 78.7 (75.9–81.6) | 7.59 |
| OHCA score | 0.844 (0.815–0.872) | <0.001 | 72.1 (68.4–75.8) | 84.2 (79.6–88.8) | 91.4 (88.8–94.0) | 56.4 (51.3–61.5) | 75.7 (72.8–78.7) | 33.98 |
| Cardiac arrest survival score | 0.831 (0.799–0.862) | <0.001 | 83.8 (80.8–86.9) | 75.9 (70.5–81.3) | 89.1 (86.4–91.7) | 66.8 (61.2–72.4) | 81.5 (78.8–84.2) | 7.0 |
| C-GRApH score | 0.771 (0.737–0.805) | <0.001 | 93.6 (91.6–95.6) | 36.1 (30.0–42.2) | 77.4 (74.2–80.5) | 70.7 (62.7–78.8) | 76.4 (73.4–79.3) | 3 |
| SR-QOLl score | 0.749 (0.711–0.786) | <0.001 | 92.4 (90.2–94.6) | 48.1 (41.8–54.4) | 80.6 (77.6–83.7) | 73.0 (66.1–79.9) | 79.1 (76.3–81.9) | 50 |

This analysis only included 804 patients for whom all of the 12 prognostication scores were available. The cut-off for PROLOGUE indicates a cut-off point of the poor outcome probability predicted using PROLOGUE. AUC, area under the receiver operating characteristic curve; CI, confidence interval; SEN, sensitivity; SPE, specificity; PPV, positive predictive value; NPV, negative predictive value; ACC, accuracy; PROLOGUE, PROgnostication using LOGistic regression model for Unselected adult cardiac arrest patients in the Early stages; TTM, targeted temperature management; CAHP, cardiac arrest hospital prognosis; rCAST, revised post-cardiac arrest syndrome for therapeutic hypothermia; PHR, Polish hypothermia registry; OHCA, out-of-hospital cardiac arrest; SR-QOLl, Serbian quality of life long-term.

Prognostication scores commonly estimate outcomes using combination of predictor variables selected through logistic regression. However, the studied scores vary greatly in terms of complexity. The prediction score by Aschauer et al. is composed of only four variables, whereas PROLOGUE is composed of 12 variables. Some scores are simply calculated as the sum of points awarded for each of the variables that are present [8–14], whereas others are calculated using complex formulas or nomograms [5–7, 15, 16]. Among those in this study, the

**Table 4. Performances of cardiac arrest-specific prognostication scores in predicting poor outcome at 6 months after cardiac arrest after multiple imputation.**

| Prediction score | AUC (95% CI) | P value | SEN (95% CI) | SPE (95% CI) | PPV (95% CI) | NPV (95% CI) | ACC (95% CI) | Cut-off |
|---|---|---|---|---|---|---|---|---|
| PROLOGUE | 0.906 (0.888–0.924) | <0.001 | 80.8 (78.0–83.5) | 85.7 (82.1–89.3) | 92.7 (90.8–94.7) | 66.4 (62.1–70.7) | 82.3 (80.1–84.5) | 0.752 |
| TTM risk score | 0.903 (0.884–0.922) | <0.001 | 89.8 (87.7–91.9) | 73.1 (68.5–77.7) | 88.3 (86.1–90.5) | 76.1 (71.6–80.6) | 84.7 (82.6–86.8) | 13 |
| CAHP score | 0.890 (0.869–0.910) | <0.001 | 81.8 (79.1–84.4) | 82.4 (78.4–86.3) | 91.3 (89.2–93.3) | 66.7 (62.3–71.1) | 81.9 (79.7–84.2) | 162.66 |
| Prediction score by Aschauer et al. | 0.885 (0.863–0.906) | <0.001 | 88.5 (86.3–90.7) | 72.8 (68.2–77.4) | 88.0 (85.8–90.3) | 73.7 (69.1–78.2) | 83.7 (81.5–85.8) | 20 |
| NULL-PLEASE score | 0.869 (0.847–0.891) | <0.001 | 90.8 (88.8–92.8) | 62.5 (57.4–67.5) | 84.5 (82.1–86.9) | 75.1 (70.2–80.0) | 82.1 (79.9–84.3) | 5 |
| 5-R score | 0.873 (0.853–0.894) | <0.001 | 75.4 (72.5–78.4) | 82.1 (78.1–86.1) | 90.5 (88.3–92.7) | 59.7 (55.3–64.0) | 77.5 (75.1–77.9) | 4 |
| rCAST score | 0.846 (0.822–0.871) | <0.001 | 77.7 (74.8–80.5) | 79.6 (75.4–83.7) | 89.6 (87.3–91.8) | 61.2 (56.8–65.6) | 78.2 (75.9–80.6) | 10.0 |
| PHR risk score | 0.857 (0.833–0.880) | <0.001 | 75.9 (73.0–78.9) | 80.1 (76.0–84.3) | 89.6 (87.3–91.9) | 59.6 (55.2–64.0) | 77.2 (74.8–79.6) | 7.59 |
| OHCA score | 0.831 (0.806–0.855) | <0.001 | 71.7 (68.6–74.8) | 82.4 (78.4–86.3) | 90.2 (87.9–92.5) | 56.3 (52.1–60.6) | 75.0 (72.5–77.5) | 33.67 |
| Cardiac arrest survival score | 0.817 (0.790–0.844) | <0.001 | 82.6 (80.0–85.2) | 74.8 (70.3–79.3) | 88.1 (85.8–90.4) | 65.6 (61.0–70.2) | 80.2 (77.9–82.5) | 7.0 |
| C-GRApH score | 0.755 (0.725–0.784) | <0.001 | 93.4 (91.7–95.1) | 35.3 (30.3–40.3) | 76.5 (73.9–79.2) | 70.4 (63.7–77.1) | 75.6 (73.1–78.0) | 3 |
| SR-QOLl score | 0.730 (0.698–0.761) | <0.001 | 91.2 (89.2–93.1) | 46.2 (41.0–51.4) | 79.3 (76.7–81.9) | 69.9 (64.1–75.8) | 77.4 (75.0–79.8) | 46.0 |

The cut-off for PROLOGUE indicates a cut-off point of the poor outcome probability predicted using PROLOGUE. AUC, area under the receiver operating characteristic curve; CI, confidence interval; SEN, sensitivity; SPE, specificity; PPV, positive predictive value; NPV, negative predictive value; ACC, accuracy; PROLOGUE, PROgnostication using LOGistic regression model for Unselected adult cardiac arrest patients in the Early stages; TTM, targeted temperature management; CAHP, cardiac arrest hospital prognosis; rCAST, revised post-cardiac arrest syndrome for therapeutic hypothermia; PHR, Polish hypothermia registry; OHCA, out-of-hospital cardiac arrest; SR-QOLl, Serbian quality of life long-term.

**Table 5. Sensitivity, specificity, positive and negative predictive values, and accuracy for different cut-offs in predicting poor outcome at 6 months after cardiac arrest.**

| Model | Cut-off | Sensitivity (95% CI) | Specificity (95% CI) | PPV (95% CI) | NPV (95% CI) | Accuracy (95% CI) |
|---|---|---|---|---|---|---|
| PROLOGUE | ≥0.1 | 98.4 (97.4–99.4) | 31.5 (25.7–37.4) | 77.1 (74.0–80.1) | 89.4 (82.9–96.0) | 78.4 (75.5–81.2) |
| | ≥0.2 | 96.3 (94.7–97.8) | 48.1 (41.8–54.4) | 81.3 (78.3–84.2) | 84.7 (78.6–90.7) | 81.8 (79.2–84.5) |
| | ≥0.4 | 94.3 (92.4–96.2) | 65.1 (59.1–71.2) | 86.3 (83.6–89.1) | 83.1 (77.7–88.4) | 85.6 (83.1–88.0) |
| | ≥0.6 | 89.9 (87.4–92.4) | 78.0 (72.8–83.2) | 90.5 (88.1–92.9) | 76.7 (71.4–82.0) | 86.3 (83.9–88.7) |
| | ≥0.8 | 78.9 (75.5–82.2) | 88.8 (84.8–92.8) | 94.3 (92.2–96.4) | 64.3 (59.1–69.4) | 81.8 (79.2–84.5) |
| | ≥0.9 | 64.3 (60.3–68.3) | 96.7 (94.4–98.9) | 97.8 (96.4–99.3) | 53.7 (49.0–58.4) | 74.0 (71.0–77.0) |
| | ≥0.95 | 51.0 (46.8–55.1) | 98.8 (97.4–100.0) | 99.0 (97.8–100.0) | 46.3 (42.0–50.6) | 65.3 (62.0–68.6) |
| TTM risk score | >10 | 95.2 (93.4–97.0) | 60.6 (54.4–66.8) | 84.9 (82.2–87.7) | 84.4 (79.0–89.8) | 84.8 (82.3–87.3) |
| | >13 | 87.6 (84.8–90.3) | 79.7 (74.6–84.7) | 91.0 (88.5–93.4) | 73.3 (67.9–78.6) | 85.2 (82.7–87.7) |
| | >16 | 71.4 (67.7–75.1) | 91.3 (87.7–94.8) | 95.0 (93.0–97.1) | 57.7 (52.8–62.7) | 77.4 (74.5–80.3) |
| CAHP score | >150 | 88.1 (85.4–90.8) | 77.2 (71.9–82.5) | 90.0 (87.5–92.5) | 73.5 (68.1–79.0) | 84.8 (82.3–87.3) |
| | >200 | 55.1 (51.0–59.2) | 95.4 (92.8–98.1) | 96.6 (94.6–98.6) | 47.6 (43.2–52.1) | 67.2 (63.9–70.4) |
| Prediction score by Aschauer et al. | >12 | 95.4 (93.6–97.1) | 56.8 (50.6–63.1) | 83.8 (80.9–86.6) | 84.0 (78.4–89.7) | 83.8 (81.3–86.4) |
| | >22 | 83.7 (80.6–86.7) | 78.8 (73.7–84.0) | 90.2 (87.7–92.8) | 67.4 (61.9–72.8) | 82.2 (79.6–84.9) |
| | >30 | 62.2 (58.2–66.2) | 92.1 (88.7–95.5) | 94.9 (92.6–97.1) | 51.0 (46.3–55.7) | 71.1 (68.0–74.3) |
| | >40 | 29.8 (26.1–33.6) | 97.9 (96.1–99.7) | 97.1 (94.6–99.6) | 37.4 (33.6–41.2) | 50.2 (46.8–53.7) |
| NULL-PLEASE score | >0 | 98.6 (97.6–99.6) | 17.0 (12.3–21.8) | 73.5 (70.4–76.7) | 83.7 (73.3–94.0) | 74.1 (71.1–77.2) |
| | >1 | 96.8 (95.3–98.3) | 35.7 (29.6–41.7) | 77.9 (74.8–80.9) | 82.7 (75.4–90.0) | 78.5 (75.6–81.3) |
| | >2 | 95.4 (93.6–97.1) | 54.4 (48.1–60.6) | 83.0 (80.1–85.9) | 83.4 (77.6–89.3) | 83.1 (80.5–85.7) |
| | >3 | 91.8 (89.6–94.1) | 65.1 (59.1–71.2) | 86.0 (83.3–88.8) | 77.3 (71.6–83.1) | 83.8 (81.3–86.4) |
| | >4 | 86.5 (83.7–89.3) | 76.3 (71.0–81.7) | 89.5 (86.9–92.1) | 70.8 (65.2–76.3) | 83.5 (80.9–86.0) |
| | >5 | 75.7 (72.1–79.2) | 85.9 (81.5–90.3) | 92.6 (90.2–95.0) | 60.2 (55.0–65.3) | 78.7 (75.9–81.6) |
| | >6 | 62.7 (58.7–66.7) | 91.3 (87.7–94.8) | 94.4 (92.1–96.7) | 51.2 (46.4–55.9) | 71.3 (68.1–74.4) |
| | >7 | 45.8 (41.7–49.9) | 96.3 (93.9–98.7) | 96.6 (94.5–98.8) | 43.2 (39.0–47.4) | 60.9 (57.6–64.3) |
| 5-R score | ≤0 | 0.7 (0–1.4) | 100.0 (100.0–100.0) | 100.0 (100.0–100.0) | 30.1 (26.9–33.3) | 30.5 (27.3–33.7) |
| | ≤1 | 14.4 (11.5–17.3) | 100.0 (100.0–100.0) | 100.0 (100.0–100.0) | 33.3 (29.9–36.8) | 40.0 (36.7–43.4) |
| | ≤2 | 41.4 (37.3–45.5) | 98.3 (96.7–100.0) | 98.3 (96.7–100.0) | 41.8 (37.7–45.9) | 58.5 (55.1–61.9) |
| | ≤3 | 58.1 (54.0–62.2) | 95.4 (92.8–98.1) | 96.7 (94.9–98.6) | 49.4 (44.8–53.9) | 69.3 (66.1–72.5) |
| | ≤4 | 75.1 (71.6–78.7) | 82.6 (77.8–87.4) | 91.0 (88.4–93.6) | 58.7 (53.5–63.9) | 77.4 (74.5–80.3) |
| | ≤5 | 93.3 (91.2–95.3) | 61.4 (55.3–67.6) | 85.0 (82.1–87.8) | 79.6 (73.8–85.4) | 83.7 (81.2–86.3) |
| | ≤6 | 95.7 (94.1–97.4) | 46.5 (40.2–52.8) | 80.7 (77.7–83.7) | 82.4 (75.9–88.8) | 81.0 (78.3–83.7) |
| rCAST score | ≥6 | 92.7 (90.6–94.9) | 48.5 (42.2–54.9) | 80.8 (77.8–83.8) | 74.1 (67.2–80.9) | 79.5 (76.7–82.3) |
| | ≥14.5 | 42.1 (38.0–46.2) | 95.9 (93.3–98.4) | 96.0 (93.5–98.4) | 41.5 (37.4–45.6) | 58.2 (54.8–61.6) |
| PHR risk score | ≥25% | 89.5 (87.0–92.1) | 58.9 (52.7–65.1) | 83.6 (80.6–86.5) | 70.6 (64.4–76.9) | 80.3 (77.6–83.1) |
| | ≥50% | 66.1 (62.2–70.0) | 87.6 (83.4–91.7) | 92.5 (90.0–95.1) | 52.5 (47.6–57.4) | 72.5 (69.4–75.6) |
| | ≥75% | 34.3 (30.4–38.2) | 96.7 (94.4–98.9) | 96.0 (93.3–98.7) | 38.6 (34.8–42.5) | 53.0 (49.5–56.4) |
| OHCA score | >2 | 98.0 (96.9–99.2) | 18.3 (13.4–23.1) | 73.7 (70.5–76.9) | 80.0 (69.4–90.6) | 74.1 (71.1–77.2) |
| | >17.4 | 91.3 (89.0–93.6) | 49.8 (43.5–56.1) | 80.9 (77.9–84.0) | 71.0 (64.2–77.8) | 78.9 (76.0–81.7) |
| | >32.5 | 74.8 (71.2–78.4) | 80.9 (76.0–85.9) | 90.1 (87.4–92.9) | 57.9 (52.6–63.1) | 76.6 (73.7–79.5) |
| Cardiac arrest survival score | ≥5 | 87.4 (84.6–90.1) | 66.4 (60.4–72.4) | 85.9 (83.0–88.7) | 69.3 (63.3–75.2) | 81.1 (78.4–83.8) |
| | ≥11 | 51.2 (47.0–55.3) | 90.0 (86.3–93.8) | 92.3 (89.4–95.3) | 44.1 (39.7–48.5) | 62.8 (59.5–66.2) |
| | ≥16 | 15.6 (12.6–18.6) | 98.3 (96.7–100.0) | 95.7 (91.5–99.8) | 33.3 (29.8–36.7) | 40.4 (37.0–43.8) |
| C-GRApH score | ≥2 | 93.6 (91.6–95.6) | 36.1 (30.0–42.2) | 77.4 (74.2–80.5) | 70.7 (62.7–78.8) | 76.4 (73.4–79.3) |
| | ≥4 | 27.0 (23.3–30.7) | 95.0 (92.3–97.8) | 92.7 (88.7–96.7) | 35.8 (32.1–39.5) | 47.4 (43.9–50.8) |

(*Continued*)

**Table 5.** (Continued)

| Model | Cut-off | Sensitivity (95% CI) | Specificity (95% CI) | PPV (95% CI) | NPV (95% CI) | Accuracy (95% CI) |
|---|---|---|---|---|---|---|
| SR-QOLl score | <25% | 29.0 (25.2–32.7) | 92.1 (88.7–95.5) | 89.6 (85.1–94.0) | 35.7 (31.9–39.5) | 47.9 (44.4–51.3) |
| | <50% | 57.4 (53.3–61.5) | 69.7 (63.9–75.5) | 81.6 (77.7–85.4) | 41.2 (36.4–46.0) | 61.1 (57.7–64.4) |
| | <75% | 80.3 (77.0–83.6) | 53.9 (47.6–60.2) | 80.3 (77.0–83.6) | 53.9 (47.6–60.2) | 72.4 (69.3–75.5) |

Cut-offs were chosen based on risk group categorization proposed in the original publications for the PROLOGUE, TTM risk, CAHP, Aschauer et al., rCAST, OHCA, cardiac arrest survival, and C-GRApH scores. For the SR-QOLl and PHR risk scores, quartiles were used as cut-offs. For the NULL-PLEASE and 5-R scores, each point was used as cut-offs. The cut-off values for PROLOGUE indicate cut-off points of the poor outcome probability predicted using PROLOGUE. CI, confidence interval; PPV, positive predictive value; NPV, negative predictive value; PROLOGUE, PROgnostication using LOGistic regression model for Unselected adult cardiac arrest patients in the Early stages; TTM, targeted temperature management; CAHP, cardiac arrest hospital prognosis; rCAST, revised post-cardiac arrest syndrome for therapeutic hypothermia; PHR, Polish hypothermia registry; OHCA, out-of-hospital cardiac arrest; SR-QOLl, Serbian quality of life long-term.

PROLOGUE, TTM risk, and CAHP scores showed outstanding predictive performance (median AUC values > 0.9), but these scores require elaborate calculations, as they use a relatively complex nomogram or multi-point scoring system with a different weight for each parameter. Although these scores are relatively complex, this would not hinder practicality for clinical use if they could be calculated electronically using a desktop calculator or mobile device.

In this study, the prediction score by Aschauer et al., OHCA score, C-GRApH score, and SR-QOLl score showed acceptable discrimination but significant miscalibration. The prediction score by Aschauer et al. and C-GRApH score overestimated the actual risk of poor outcome at extreme predicted probability values, whereas the OHCA score and SR-QOLl score underestimated it. Although the calibration performances of the prediction score by Aschauer et al., C-GRApH score, and SR-QOLl score, to the best of our knowledge, have not been evaluated in separate studies, the low calibration capacity of the OHCA score for predicting poor outcome (CPC 3–5) at 6 months after OHCA has also been reported by other researchers [9, 19]. Our study suggests that these scores need to be updated for use in settings similar to ours.

These scores would allow treating physicians to provide a patient's likely long-term outcome in a more objective manner in the early hours after OHCA. Although the prognostication scores in the present study could predict poor outcome with statistical significance, they were not specific enough to be used for important therapeutic decision-making (e.g., withholding or withdrawing life-saving treatment). These scores can be used as an adjunct to guide therapeutic decision-making. However, given the insufficient specificities observed in this study, important therapeutic decisions should not be made based on these prognostication scores alone.

Our study has several limitations. First, it was a retrospective analysis of data collected from teaching hospitals in the Republic of Korea. The performances of prognostication scores may be different in other healthcare or country settings. Second, we evaluated the performances of prognostication scores, but could not assess their clinical usefulness. Further studies are required to evaluate this. Third, we could not evaluate several cardiac arrest-specific prognostication scores that required variables unavailable from our registry data [19, 30–32]. Lastly, the treating physicians were not blinded to the constituent results of the prognostication scores, thereby introducing the potential for self-fulfilling prophecy bias.

## Conclusions

We evaluated the performances of 12 existing cardiac arrest-specific prognostication scores in predicting poor outcome at 6 months after OHCA using data from a multicenter registry of

**Table 6. Pairwise comparisons of areas under the receiver operating characteristic curves.**

| | Difference between AUC (95% CI) | | | | | | | | | | |
|---|---|---|---|---|---|---|---|---|---|---|---|
| | PROLOGUE | TTM risk score | CAHP score | Prediction score by Aschauer et al. | NULL-PLEASE score | 5-R score | rCAST score | PHR risk score | OHCA score | Cardiac arrest survival score | C-GRApH score |
| TTM risk score | 0.009 (−0.004–0.023) | - | - | - | - | - | - | - | - | - | - |
| CAHP score | 0.016 (−0.003–0.036) | 0.007 (−0.010–0.024) | - | - | - | - | - | - | - | - | - |
| Prediction score by Aschauer et al. | 0.031 (0.009–0.053)* | 0.021 (0.003–0.040)* | 0.015 (−0.003–0.032) | - | - | - | - | - | - | - | - |
| NULL-PLEASE score | 0.037 (0.015–0.059)* | 0.027 (0.006–0.049)* | 0.021 (0.000–0.041)* | 0.006 (−0.018–0.030) | - | - | - | - | - | - | - |
| 5-R score | 0.044 (0.013–0.074)* | 0.034 (0.002–0.066)* | 0.027 (−0.005–0.060) | 0.013 (−0.022–0.047) | 0.007 (−0.027–0.041) | - | - | - | - | - | - |
| rCAST score | 0.055 (0.035–0.075)* | 0.046 (0.023–0.069)* | 0.039 (0.017–0.061)* | 0.024 (−0.005–0.054) | 0.018 (−0.005–0.042) | 0.012 (−0.024–0.048) | - | - | - | - | - |
| PHR risk score | 0.057 (0.032–0.082)* | 0.048 (0.026–0.069)* | 0.041 (0.018–0.064)* | 0.026 (0.004–0.049)* | 0.020 (−0.003–0.043) | 0.013 (−0.023–0.050) | 0.002 (−0.028–0.032) | - | - | - | - |
| OHCA score | 0.079 (0.053–0.104)* | 0.069 (0.042–0.097)* | 0.062 (0.039–0.086)* | 0.048 (0.021–0.075)* | 0.042 (0.017–0.067)* | 0.035 (−0.002–0.072) | 0.023 (−0.004–0.051) | 0.022 (−0.009–0.053) | - | - | - |
| Cardiac arrest survival score | 0.092 (0.062–0.122)* | 0.082 (0.056–0.109)* | 0.075 (0.046–0.105)* | 0.061 (0.032–0.090)* | 0.055 (0.033–0.077)* | 0.048 (0.009–0.088)* | 0.036 (0.002–0.071)* | 0.035 (0.011–0.058)* | 0.013 (−0.022–0.048) | - | - |
| C-GRApH score | 0.152 (0.121–0.182)* | 0.142 (0.114–0.170)* | 0.135 (0.105–0.166)* | 0.121 (0.089–0.153)* | 0.115 (0.081–0.148)* | 0.108 (0.066–0.150)* | 0.096 (0.062–0.130)* | 0.095 (0.059–0.130)* | 0.073 (0.035–0.110)* | 0.060 (0.023–0.097)* | - |
| SR-QOLl score | 0.174 (0.132–0.216)* | 0.164 (0.121–0.208)* | 0.158 (0.114–0.201)* | 0.143 (0.098–0.188)* | 0.137 (0.092–0.182)* | 0.130 (0.086–0.175)* | 0.119 (0.072–0.165)* | 0.117 (0.070–0.163)* | 0.095 (0.048–0.142)* | 0.082 (0.033–0.131)* | 0.022 (−0.029–0.073) |

This analysis only included the 804 patients for whom all of the 12 prognostication scores were available.

* $P < 0.05$ by DeLong test. AUC, area under the receiver operating characteristic curve; CI, confidence interval; PROLOGUE, PROgnostication using LOGistic regression model for Unselected adult cardiac arrest patients in the Early stages; TTM, targeted temperature management; CAHP, cardiac arrest hospital prognosis; rCAST, revised post-cardiac arrest syndrome for therapeutic hypothermia; PHR, Polish hypothermia registry; OHCA, out-of-hospital cardiac arrest; SR-QOLl, Serbian quality of life long-term.

comatose OHCA patients who underwent TTM. The PROLOGUE, TTM risk, CAHP, NULL-PLEASE, 5-R, and cardiac arrest survival scores showed satisfactory discrimination and calibration performances. Although the calibration performance was not perfect, the rCAST and PHR risk scores also showed acceptable overall calibration and good discrimination performances. The prediction score by Aschauer et al., OHCA score, C-GRApH score, and SR-QOLl score showed acceptable discrimination but significant miscalibration. None of the prognostication scores in this study were specific enough to be used alone in important therapeutic decision-making. These study findings may improve our understanding of these prognostication scores and thereby aid in the interpretations of the prediction results.

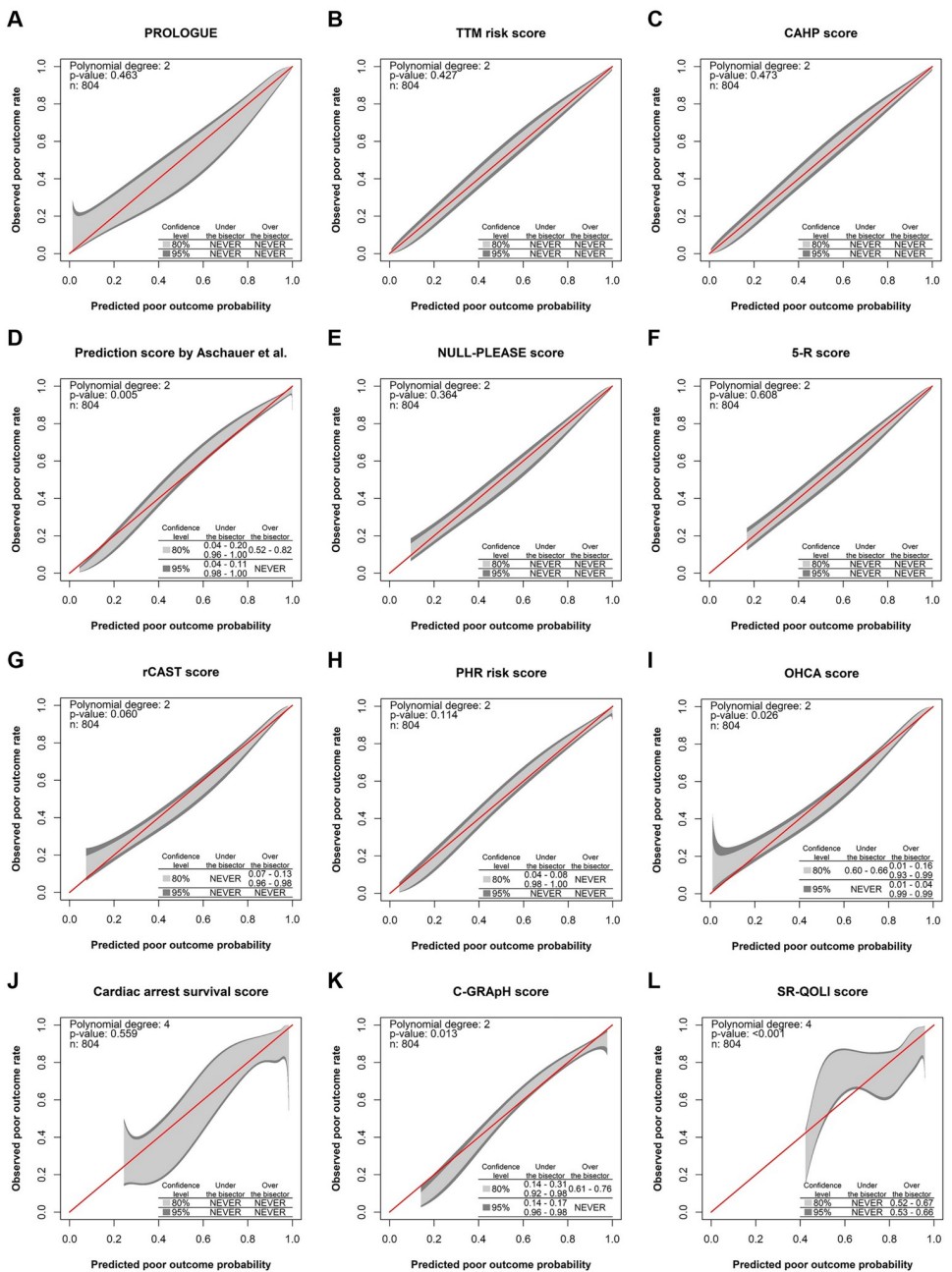

**Fig 4. Calibration belts for the prognostication scores.** (A) PROLOGUE, (B) TTM risk score, (C) CAHP score, (D) prediction score by Aschauer et al., (E) NULL-PLEASE score, (F) 5-R score, (G) rCAST score, (H) PHR risk score, (I) OHCA score, (J) cardiac arrest survival score, (K) C-GRApH score, (L) SR-QOLl score. The bisecting lines correspond to perfect agreement between observed outcomes and predicted outcomes (perfect calibration). The light and dark shaded areas represent 80% and 95% confidence intervals, respectively. This analysis only included the 804 patients for whom all of the 12 prognostication scores were available. PROLOGUE, PROgnostication using LOGistic regression model for Unselected adult cardiac arrest patients in the Early stages; TTM, targeted temperature management; CAHP, cardiac arrest hospital prognosis; rCAST, revised post-cardiac arrest syndrome for therapeutic hypothermia; PHR, Polish hypothermia registry; OHCA, out-of-hospital cardiac arrest; SR-QOLl, Serbian quality of life long-term.

## Supporting information

**S1 Data. Raw data.**
(XLSX)

## Acknowledgments

We would like to thank Editage (www.editage.co.kr) for English language editing. The following investigators participated in the Korean Hypothermia Network. Chair: Kyung Woon Jeung (Chonnam National University Hospital, E-mail: neoneti@hanmail.net). Principal investigators of each hospital: Kyu Nam Park (The Catholic University of Korea, Seoul St. Mary's Hospital); Minjung Kathy Chae (Ajou University Medical Center); Won Young Kim (Asan Medical Center); Byung Kook Lee (Chonnam National University Hospital); Dong Hoon Lee (Chung-Ang University Hospital); Tae Chang Jang (Daegu Catholic University Medical Center); Jae Hoon Lee (Dong-A University Hospital); Yoon Hee Choi (Ewha Womans University Mokdong Hospital); Je Sung You (Gangnam Severance Hospital); Young Hwan Lee (Hallym University Sacred Heart Hospital); In Soo Cho (Hanil General Hospital); Su Jin Kim (Korea University Anam Hospital); Jong-Seok Lee (Kyung Hee University Medical Center); Yong Hwan Kim (Samsung Changwon Hospital); Min Seob Sim (Samsung Medical Center); Jonghwan Shin (Seoul Metropolitan Government Seoul National University Boramae Medical Center); Yoo Seok Park (Severance Hospital); Hyung Jun Moon (Soonchunhyang University Hospital Cheonan); Won Jung Jeong (The Catholic University of Korea, St. Vincent's Hospital); Joo Suk Oh (The Catholic University of Korea, Uijeongbu St. Mary's Hospital); Seung Pill Choi (The Catholic University of Korea, Yeouido St. Mary's Hospital); Kyoung-Chul Cha (Wonju Severance Christian Hospital).

## Author Contributions

**Conceptualization:** Kyung Woon Jeung.

**Data curation:** Yong Hun Jung, Hyoung Youn Lee.

**Formal analysis:** Byung Kook Lee, Chun Song Youn.

**Funding acquisition:** Kyung Woon Jeung.

**Investigation:** Wan Young Heo, Kyung Woon Jeung, Seung Pill Choi.

**Project administration:** Kyu Nam Park, Yong Il Min.

**Supervision:** Kyu Nam Park, Yong Il Min.

**Writing – original draft:** Wan Young Heo, Kyung Woon Jeung.

**Writing – review & editing:** Yong Hun Jung, Hyoung Youn Lee, Byung Kook Lee, Chun Song Youn, Seung Pill Choi, Kyu Nam Park, Yong Il Min.

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
