## [Decision Letter · Decision Letter 0]

19 Jan 2022

PONE-D-21-38998External validation of cardiac arrest-specific prognostication scores developed for early prognosis estimation after out-of-hospital cardiac arrest in a Korean multicenter cohortPLOS ONE

Dear Dr. Jeung,

Thank you for submitting your manuscript to PLOS ONE. After careful consideration, we feel that it has merit but does not fully meet PLOS ONE’s publication criteria as it currently stands. Therefore, we invite you to submit a revised version of the manuscript that addresses the points raised during the review process.

We look forward to receiving your revised manuscript.

Kind regards,

Muhammad Tarek Abdel Ghafar, M.D

Academic Editor

PLOS ONE

Journal Requirements:

2. One of the noted authors is a group or consortium [Korean Hypothermia Network investigators]. In addition to naming the author group, please list the individual authors and affiliations within this group in the acknowledgments section of your manuscript. Please also indicate clearly a lead author for this group along with a contact email address.

Reviewers' comments:

Reviewer's Responses to Questions

**Comments to the Author**

1. Is the manuscript technically sound, and do the data support the conclusions?

Reviewer #1: Yes

Reviewer #2: Yes

Reviewer #3: Yes

Reviewer #4: Yes

2. Has the statistical analysis been performed appropriately and rigorously? 

Reviewer #1: Yes

Reviewer #2: Yes

Reviewer #3: Yes

Reviewer #4: Yes

3. Have the authors made all data underlying the findings in their manuscript fully available?

Reviewer #1: Yes

Reviewer #2: Yes

Reviewer #3: Yes

Reviewer #4: Yes

4. Is the manuscript presented in an intelligible fashion and written in standard English?

Reviewer #1: Yes

Reviewer #2: Yes

Reviewer #3: Yes

Reviewer #4: Yes

5. Review Comments to the Author

Reviewer #1: Heo et al. have undertaken a formidable task of validating scoring methods for prognostication of 6-months outcome in patients who suferred an out-of-hospital cardiac arrest. They took as an external validating database the collaborative Korean results, including nearly 1,200 subjects. They rightfully claim that a scoring method that was formed on a given group of people, does not neccessarily serves as well on a group with different characteristics, ethnic, socioeconomic and otherwise.

They used very strict inclusion/exclusion criteria, nicely explaining that all Korean patients were subjected to treament under hypothermia and excluding a group within their Korean dataset, who were used to form one of the scoring methods.

The statistical approach is generous, in the sense that it gives not only numerical data in tables but uses two graphical demonstration modes, most notbly ROC curves. Indeed, my eyes tell me that appart from 2-3 methods, all others are quite accurate. The minute differnces among them might be applicable in certain specific circumstances when facing families. I would advise to turn the ROC figure into colored one, because as it is it is not legible enough.

This manuscript is a useful adjunct for medical staff who face families near ICUs and have to reflect honestly to them their beloved one's situation and prognosis.

Reviewer #2: Dear Authors

I received your paper as a reviewer. I found that you tried to compare some scoring systems in terms of predicting outcome of OHCA cases. You had a proper data from considerable number of patients and this is an important positive point of your study.

I just appreciate your high quality study and proper presentation. I have nothing to add and vote for accept.

As a recommendation, using your valuable database you can even pooled the data and can develop a new scoring system, that have even higher accuracy compare with the previous ones.

Good luck

Reviewer #3: Appreciating your work, I would like to forward the following points:

1. Can the authors explain why the have forgone an ethical disclaimer segment in their manuscript? I believe the issue of clearance, consent, and competing interest statements as it pertains to the funding of this particular study must be explained further.

2. Can the authors please provide a statement on their conclusion that highlights the impact of their findings? I believe that would strengthen the conclusion.

Reviewer #4: Authors report an interesting study evaluating "the performance of cardiac arrest-specific prognostication scores developed for outcome prediction in the early hours after out-of-hospital cardiac arrest

(OHCA) in predicting long-term outcomes using independent data." The scores analyzed are OHCA, CAHP, C-GRApH,TTM risk, 5-R, NULL-PLEASE, SR-QOLl, cardiac arrest survival, rCAST, PHR risk, and PROLOGUE scores and prediction score by Aschauer et al.

The main results are:

- PROLOGUE score showed the highest AUC

- SR-QOLl score had the lowest AUC.

- PROLOGUE, TTM risk, CAHP, NULLPLEASE, 5-R, and cardiac arrest survival scores were well calibrated.

- rCAST and PHR risk scores showed acceptable overall calibration, although they showed

miscalibration under the 80% CI level at extreme prediction values.

- OHCA score,C-GRApH score, prediction score by Aschauer et al., and SR-QOLl score showed

significant miscalibration

The authors are cautious about their results and recognized some important limitations. as the retrospective type of the study and low specificity of each score. These tests cannot be used alone to guide OHCA treatment.

Here are my systematic comments

1- Abstract

Abbreviations must be defined at the first time.

No other comments

2- Introduction

No major comments. Are all the reported studies retrospective?

3- Methods

The methods are well designed. in Table 1 authors must add references for each score.

4- Results

No comments

5- Discussion

No comments. Limitations are well analyzed.

6- Conclusion

No specific comments

6. PLOS authors have the option to publish the peer review history of their article (what does this mean?). If published, this will include your full peer review and any attached files.

Reviewer #1: **Yes: **‪Izhar Ben-Shlomo‬‏, MD, Head of Emergency Medicine program, Zefat Academic College

Reviewer #2: **Yes: **Alireza Baratloo

Reviewer #3: No

Reviewer #4: No

---

## [Author Response · Author response to Decision Letter 0]

25 Jan 2022

Dear editor,

Firstly, we appreciate you for your comments. They were very helpful in improving our manuscript and included very useful points that we had not previously recognized. After due consideration, the manuscript was revised as described below.

Journal Requirements:

: Our manuscript was changed to meet PLOS ONE’s style requirements.

2. One of the noted authors is a group or consortium [Korean Hypothermia Network investigators]. In addition to naming the author group, please list the individual authors and affiliations within this group in the acknowledgments section of your manuscript. Please also indicate clearly a lead author for this group along with a contact email address.

: Information on network chair and principal investigators of each participating hospital was added in Acknowledgments section as below.

The following investigators participated in the Korean Hypothermia Network. Chair: Kyung Woon Jeung (Chonnam National University Hospital, E-mail: neoneti@hanmail.net). Principal investigators of each hospital: Kyu Nam Park (The Catholic University of Korea, Seoul St. Mary’s Hospital); Minjung Kathy Chae (Ajou University Medical Center); Won Young Kim (Asan Medical Center); Byung Kook Lee (Chonnam National University Hospital); Dong Hoon Lee (Chung-Ang University Hospital); Tae Chang Jang (Daegu Catholic University Medical Center); Jae Hoon Lee (Dong-A University Hospital); Yoon Hee Choi (Ewha Womans University Mokdong Hospital); Je Sung You (Gangnam Severance Hospital); Young Hwan Lee (Hallym University Sacred Heart Hospital); In Soo Cho (Hanil General Hospital); Su Jin Kim (Korea University Anam Hospital); Jong-Seok Lee (Kyung Hee University Medical Center); Yong Hwan Kim (Samsung Changwon Hospital); Min Seob Sim (Samsung Medical Center); Jonghwan Shin (Seoul Metropolitan Government Seoul National University Boramae Medical Center); Yoo Seok Park (Severance Hospital); Hyung Jun Moon (Soonchunhyang University Hospital Cheonan); Won Jung Jeong (The Catholic University of Korea, St. Vincent’s Hospital); Joo Suk Oh (The Catholic University of Korea, Uijeongbu St. Mary’s Hospital); Seung Pill Choi (The Catholic University of Korea, Yeouido St. Mary’s Hospital); Kyoung-Chul Cha (Wonju Severance Christian Hospital).

: We reviewed the references in the list. There was no retracted article among the cited papers.

Thank you again for your invaluable suggestions for improving our manuscript.

Sincerely,

Dear reviewer #1,

Firstly, we appreciate you for your comments. They were very helpful in improving our manuscript and included very useful points that we had not previously recognized. After due consideration, the manuscript was revised as described below.

Reviewer #1: Heo et al. have undertaken a formidable task of validating scoring methods for prognostication of 6-months outcome in patients who suffered an out-of-hospital cardiac arrest. They took as an external validating database the collaborative Korean results, including nearly 1,200 subjects. They rightfully claim that a scoring method that was formed on a given group of people, does not neccessarily serves as well on a group with different characteristics, ethnic, socioeconomic and otherwise.

They used very strict inclusion/exclusion criteria, nicely explaining that all Korean patients were subjected to treatment under hypothermia and excluding a group within their Korean dataset, who were used to form one of the scoring methods.

The statistical approach is generous, in the sense that it gives not only numerical data in tables but uses two graphical demonstration modes, most notbly ROC curves. Indeed, my eyes tell me that appart from 2-3 methods, all others are quite accurate. The minute differences among them might be applicable in certain specific circumstances when facing families. I would advise to turn the ROC figure into colored one, because as it is it is not legible enough.

This manuscript is a useful adjunct for medical staff who face families near ICUs and have to reflect honestly to them their beloved one's situation and prognosis.

: To improve readability, the Fig. 3 was modified with colored and dashed lines.

Thank you again for your invaluable suggestions for improving our manuscript.

Sincerely,

Dear reviewer #2,

Firstly, we appreciate you for your comments. They were very helpful in improving our manuscript and included very useful points that we had not previously recognized. After due consideration, the manuscript was revised as described below.

Reviewer #2: Dear Authors

I received your paper as a reviewer. I found that you tried to compare some scoring systems in terms of predicting outcome of OHCA cases. You had a proper data from considerable number of patients and this is an important positive point of your study.

I just appreciate your high quality study and proper presentation. I have nothing to add and vote for accept.

As a recommendation, using your valuable database you can even pooled the data and can develop a new scoring system, that have even higher accuracy compare with the previous ones.

Good luck

: Thank you for this advice. We will consider it as a fascinating future study.

Thank you again for your invaluable suggestions for improving our manuscript.

Sincerely,

Dear reviewer #3,

Firstly, we appreciate you for your comments. They were very helpful in improving our manuscript and included very useful points that we had not previously recognized. After due consideration, the manuscript was revised as described below.

Reviewer #3: Appreciating your work, I would like to forward the following points:

1. Can the authors explain why the have forgone an ethical disclaimer segment in their manuscript? I believe the issue of clearance, consent, and competing interest statements as it pertains to the funding of this particular study must be explained further.

: This study conformed to the principles outlined in the Declaration of Helsinki. It was a retrospective analysis of data from the Korean Hypothermia Network prospective registry, which enrolled adult OHCA patients treated with targeted temperature management at 22 teaching hospitals in the Republic of Korea. The study design and registry protocol were approved by the institutional review board of all participating hospitals, including the Chonnam National University Hospital Institutional Review Board (CNUH-2015-164), and registered at the International Clinical Trials Registry Platform (ClinicalTrials.gov identifier: NCT02827422). In accordance with national requirements and the principles of the Declaration of Helsinki, written informed consent was obtained from all patients' legal surrogates. The following changes were made to explain these further.

The sentence “This study conformed to the principles outlined in the Declaration of Helsinki.” was added to the method section.

The sentence “In brief, a principal investigator at each participating hospital reviewed the medical records of patients who were eligible for enrollment in the registry and collected demographic, prehospital resuscitation, in-hospital treatment, and outcome data using a web-based case report form based on the Utstein Resuscitation Registry Templates [23].” in the method section was changed to “In brief, a principal investigator at each participating hospital reviewed the medical records of patients who were eligible for registry enrollment and collected their demographic, prehospital resuscitation, in-hospital treatment, and outcomes data in an anonymous fashion using a web-based case report form based on the Utstein Resuscitation Registry Templates [23].”. 

The sentence “Collection and analysis of the registry data were approved by the Institutional Review Board of each participating hospital, and written informed consent was obtained from the patients’ legal surrogates for all patients enrolled in the registry.” in the manuscript and ethics statement section of the submission system was changed to “The study design and registry protocol were approved by the institutional review board of all participating hospitals, including the Chonnam National University Hospital Institutional Review Board (CNUH-2015-164) and registered at the International Clinical Trials Registry Platform (ClinicalTrials.gov identifier: NCT02827422). Written informed consent was obtained from the legal surrogates of all patients enrolled in the registry.”.

This study was supported by a grant (BCRI21040) from the Chonnam National University Hospital Biomedical Research Institute (Recipient: KWJ). The funders had no role in the design of the study; in the collection, analyses, or interpretation of data; in the writing of the manuscript; or in the decision to publish the results. All authors have no known competing financial interests or personal relationships that could have appeared to influence the work reported in this paper. According to the submission guidelines of the PLOS ONE journal, the financial disclosure and competing interests statements were not included in the manuscript, but inserted in the financial disclosure and competing interests sections of the submission system, respectively.

2. Can the authors please provide a statement on their conclusion that highlights the impact of their findings? I believe that would strengthen the conclusion.

: To highlight the impact of our findings, the sentence “These study findings may improve our understanding of these prognostication scores and thereby aid in the interpretations of the prediction results.” was added to the conclusions section.

Thank you again for your invaluable suggestions for improving our manuscript.

Sincerely,

Dear reviewer #4,

Firstly, we appreciate you for your comments. They were very helpful in improving our manuscript and included very useful points that we had not previously recognized. After due consideration, the manuscript was revised as described below.

Reviewer #4: Authors report an interesting study evaluating "the performance of cardiac arrest-specific prognostication scores developed for outcome prediction in the early hours after out-of-hospital cardiac arrest (OHCA) in predicting long-term outcomes using independent data." The scores analyzed are OHCA, CAHP, C-GRApH, TTM risk, 5-R, NULL-PLEASE, SR-QOLl, cardiac arrest survival, rCAST, PHR risk, and PROLOGUE scores and prediction score by Aschauer et al.

The main results are:

- PROLOGUE score showed the highest AUC

- SR-QOLl score had the lowest AUC.

- PROLOGUE, TTM risk, CAHP, NULLPLEASE, 5-R, and cardiac arrest survival scores were well calibrated.

- rCAST and PHR risk scores showed acceptable overall calibration, although they showed

miscalibration under the 80% CI level at extreme prediction values.

- OHCA score, C-GRApH score, prediction score by Aschauer et al., and SR-QOLl score showed

significant miscalibration

The authors are cautious about their results and recognized some important limitations. as the retrospective type of the study and low specificity of each score. These tests cannot be used alone to guide OHCA treatment.

Here are my systematic comments

1- Abstract

Abbreviations must be defined at the first time.

No other comments

: The sentence “The following scores were calculated for 1,163 OHCA patients who were treated with targeted temperature management at 21 hospitals in South Korea: OHCA, CAHP, C-GRApH, TTM risk, 5-R, NULL-PLEASE, SR-QOLl, cardiac arrest survival, rCAST, PHR risk, and PROLOGUE scores and prediction score by Aschauer et al.” was changed to “The following scores were calculated for 1,163 OHCA patients who were treated with targeted temperature management (TTM) at 21 hospitals in South Korea: OHCA, cardiac arrest hospital prognosis (CAHP), C-GRApH (named on the basis of its variables), TTM risk, 5-R, NULL-PLEASE (named on the basis of its variables), Serbian quality of life long-term (SR-QOLl), cardiac arrest survival, revised post-cardiac arrest syndrome for therapeutic hypothermia (rCAST), Polish hypothermia registry (PHR) risk, and PROgnostication using LOGistic regression model for Unselected adult cardiac arrest patients in the Early stages (PROLOGUE) scores and prediction score by Aschauer et al.”.

2- Introduction

No major comments. Are all the reported studies retrospective?

: Most scores were derived from retrospective studies, but the OHCA and SR-QOLl scores were from prospective studies. To clarify this, the sentence “Several cardiac arrest-specific prognostication scores based on variables readily available at hospital admission have been introduced for use in the early hours after OHCA [5–16].” was changed to “Several cardiac arrest–specific prognostication scores for use in the early hours after OHCA have been developed from retrospective or prospective analyses of OHCA data [5–16].”.

Most of the existing validation studies were retrospective studies, whereas the study by Luescher et al. (reference no. 18) was a prospective study. To clarify this, the sentence “External validations in various patient populations are thus required to enable widespread reliance on a risk prediction score, but only few such scores have undergone any external validation using independent data; where this has been done, it is usually limited to discrimination performance analysis [7,9,17–22].” was changed to “Thus, external validations in various patient populations are required to enable widespread reliance on a risk prediction score, but few such scores have undergone any external validation using independent data; where this has been done, it was usually limited to retrospective analyses of discrimination performance [7,9,17–22].”.

3- Methods

The methods are well designed. in Table 1 authors must add references for each score.

: The corresponding reference number for each score was added in Table 1.

4- Results

No comments

5- Discussion

No comments. Limitations are well analyzed.

6- Conclusion

No specific comments

The following changes were made in addition to the above changes.

The term “Servian” was changed to “Serbian” throughout the manuscript.

The expression “named for its variables” was changed to “named on the basis of its variables”.

Thank you again for your invaluable suggestions for improving our manuscript.

Sincerely,

---

## [Decision Letter · Decision Letter 1]

28 Feb 2022

External validation of cardiac arrest-specific prognostication scores developed for early prognosis estimation after out-of-hospital cardiac arrest in a Korean multicenter cohort

PONE-D-21-38998R1

Dear Dr. Jeung,

We’re pleased to inform you that your manuscript has been judged scientifically suitable for publication and will be formally accepted for publication once it meets all outstanding technical requirements.

Kind regards,

Muhammad Tarek Abdel Ghafar, M.D

Academic Editor

PLOS ONE

Additional Editor Comments (optional):

Reviewers' comments:

Reviewer's Responses to Questions

**Comments to the Author**

1. If the authors have adequately addressed your comments raised in a previous round of review and you feel that this manuscript is now acceptable for publication, you may indicate that here to bypass the “Comments to the Author” section, enter your conflict of interest statement in the “Confidential to Editor” section, and submit your "Accept" recommendation.

Reviewer #3: All comments have been addressed

Reviewer #4: All comments have been addressed

2. Is the manuscript technically sound, and do the data support the conclusions?

Reviewer #3: Yes

Reviewer #4: Yes

3. Has the statistical analysis been performed appropriately and rigorously? 

Reviewer #3: Yes

Reviewer #4: Yes

4. Have the authors made all data underlying the findings in their manuscript fully available?

Reviewer #3: Yes

Reviewer #4: Yes

5. Is the manuscript presented in an intelligible fashion and written in standard English?

Reviewer #3: Yes

Reviewer #4: Yes

6. Review Comments to the Author

Reviewer #3: I would like to thank the authors for taking the feedback towards the betterment of the manuscript. All prior questions have been addressed. No further comments.

Reviewer #4: I do not have any new comments. The authors took into account all of my comments and suggestions. This research can help to improve outcomes of OHCA.

7. PLOS authors have the option to publish the peer review history of their article (what does this mean?). If published, this will include your full peer review and any attached files.

Reviewer #3: No

Reviewer #4: No

---

## [Editor Report · Acceptance letter]

23 Mar 2022

PONE-D-21-38998R1 

External validation of cardiac arrest-specific prognostication scores developed for early prognosis estimation after out-of-hospital cardiac arrest in a Korean multicenter cohort 

Dear Dr. Jeung:

I'm pleased to inform you that your manuscript has been deemed suitable for publication in PLOS ONE. Congratulations! Your manuscript is now with our production department. 

Kind regards, 

on behalf of

Prof Muhammad Tarek Abdel Ghafar 

Academic Editor

PLOS ONE